# Comparative iTRAQ proteomic profiling of proteins associated with the adaptation of brown planthopper to moderately resistant vs. susceptible rice varieties

Wenjun Zha[1], Aiqing You[1,2]*

**1** Hubei Key Laboratory of Food Crop Germplasm and Genetic Improvement, Food Crops Institute, Hubei Academy of Agricultural Sciences, Wuhan, China, **2** Hubei Collaborative Innovation Center for Grain Industry, Yangtze University, Jingzhou, China

* aq_you@163.com

**Data Availability Statement:** All relevant data are within the manuscript and its Supporting Information files.

**Funding:** Financial support was provided by the Major Program of Genetically Modified Organisms

## Abstract

The brown planthopper (BPH), *Nilaparvata lugens* (Stål), is a destructive pest that poses a significant threat to rice plants worldwide. To explore how BPHs adapt to the resistant rice variety, we analyzed proteomics profiles of two virulent *N. lugens* populations. We focused on Biotype Y, which can survive on the moderately resistant rice variety YHY15, and Biotype I, which can survive on the susceptible rice variety TN1. We performed protein quantitation using the isobaric tag for relative and absolute quantification (iTRAQ) and then compared the expression patterns between two virulent *N. lugens* populations and found 258 differentially expressed proteins (DEPs). We found that 151 of the DEPs were up-regulated, while 107 were down-regulated. We evaluated transcript levels of 8 expressed genes from the iTRAQ results by qRT-PCR, which revealed transcriptional changes that were consistent with the changes at the protein level. The determination of the protein changes in two virulent *N. lugens* populations would help to better understanding BPH adaptation to resistant rice varieties and facilitate the better design of new control strategies for host defense against BPH.

## Introduction

Proteomics is the study of proteins and their impact on cells, tissues and organisms. The data obtained from proteomic analyses provide a macroscopic view of gene expression under different environmental conditions, thereby enabling targeted experiments. Traditionally, proteomic studies based on two-dimensional electrophoresis were used to identify differentially expressed proteins (DEPs), including those related to rapid cold stress response in *Lissorhoptrus oryzophilus* [1] and immune response in *Triatoma dimidiate* [2]. With recent technological advances, highly sensitive proteomics techniques have emerged. These new approaches include the isobaric tag for relative and absolute quantification (iTRAQ), which is more reliable than two-dimensional electrophoresis [3]. This high-throughput approach enables DEPs

Breeding of China (2018ZX08001-01B), the
National Key Research and Development Program
of China (2018YFD0100806) to WZ, and the
Supportive Project of Hubei Academy of
Agricultural Sciences (2019fcxjh02) to WZ.

**Competing interests:** The authors have declared
that no competing interests exist.

to be incorporated into pathway models [4]. Recent research has shown that iTRAQ can be utilized in many different species, including *Laodelphax striatellus* [5], *Autographa californica* [6] and *Apis mellifera* [7]. A previous study of brown planthopper (*Nilaparvata lugens* Stål) used an iTRAQ protocol to identify proteins present in the ovary [8].

The brown planthopper is a destructive rice pest that causes severe damage and results in significant yield loss through both direct grain loss as well as transmission of plant viruses [9]. Although insecticides can be used to control damage from BPH and other pests, over usage of them has led to BPH resurgence [10] and has caused environmental problems that threaten human health. Since the first resistant rice variety against BPH was discovered in 1969, more than 30 BPH resistance genes have been reported from different resistance sources [11]. We used a susceptible rice variety (TN1) as a control and a moderately resistant rice variety (YHY15) carrying the resistance gene BPH15 [12]. It has been found that resistance genes impair BPH feeding behavior on varieties and cause BPH physiological changes by increasing mortality rates, extending developmental periods, and reducing reproductive output [10, 13, 14]. BPHs that are allowed to feed on resistant rice for a long time may slowly evolve into new virulent BPH populations to overcome rice resistance [15]. Among different brown planthopper biotypes, the BPH biotype I is widely distributed in East and Southeast Asia and can survive on the TN1 rice variety [16]. The BPH biotype Y is a virulent biotype that has adapted to the moderately resistant rice variety (YHY15) by compelling biotype I BPHs to feed on YHY15 for generations [17]. In both lab and field studies, brown planthopper has been able to adapt to resistant rice rapidly.

In this study, iTRAQ was used to evaluate proteomic differences between two BPH populations, leading to the identification of DEPs, which are correlated with resistance. Among 3167 identified proteins, 258 were considered as differentially expressed in the BPH-YHY15 population relative to the BPH-TN1 population. We then used Gene Ontology (GO) annotations and Kyoto Encyclopedia of Genes and Genomes (KEGG) pathway analysis to analyze the functions of these DEPs. Subsequent Clusters of Orthologous Groups of proteins (COG) analysis suggested that a number of those proteins were involved in the regulation of post-translational modification, protein turnover, chaperones pathways. Additional research on these proteins might provide valuable information regarding strategies for BPH management and control.

## Materials and methods

### Plants and insects

For this study, we utilized TN1 as a susceptible rice variety and YHY15 as a moderately resistant rice variety. Two virulent *N. lugens* populations (Biotype I and Biotype Y) were maintained on TN1 and YHY15 plants at the Hubei Academy of Agricultural Sciences, Wuhan, China.

### Protein extraction

Three different replicates were carried out, using forty adult insects in each. The phenol/methanolic ammonium acetate method was used for protein extraction [18]. The samples were dissolved in extraction media and suspended for 5 s, broken by ultrasonication (2.4 s on, 3 s off) and then incubated at 22˚C for 20 min before final centrifugation at $14,000 \times g$ for 25 min at 4˚C. The extract containing total insect protein was kept at -80˚C for further quantitative proteomic analysis.

## iTRAQ labeling and fractionation

Protein digestion was conducted as described previously [18–20]. An aliquot of 200 μg of total protein was collected from each sample and digested with 2 μg trypsin (Promega). Samples were labeled according to the procedure outlined in the 4-plex iTRAQ reagent (Applied Biosystems). Protein samples were labeled as TN1-1:115; YHY15-1:117; TN1-2:115; YHY15-2:117; TN1-3:116; YHY15-3:114, followed by multiplexing and vacuum drying. The labeled samples were collected using a chromatography system and then separated into ten fractions via a substantial cation exchange column.

## Nano-LC-MS/MS analysis

Peptides from each fraction were dissolved in solution A (5% ACN, 0.1% FA) and then centrifuged. The supernatant was then fractionated using a RIGOL L-3000RP-HPLC system. Elution was performed by transitioning from 5% solution B (95% ACN, 0.1% FA) to 35% solution B over 40 min, followed by a 5 min linear gradient from 60% up to 80% over 2 min and then 1 min of constant running at 80%. Finally, a 5% solution was used for 1 min and the instrument was equilibrated in solution A. Nano-LC-MS/MS analysis was carried out using a nano-LC equipped with a Q Exactive Mass Spectrometer.

Reference proteins were obtained from the Hemiptera protein database downloaded from the National Center for Biotechnology Information database. Proteins with fold-change values $\geq 1.2$ and $p$-values $\leq 0.05$ were considered to be significant DEPs between the two samples.

## Bioinformatics analysis

Protein functional annotation of the identified DEPs was carried out using Uniprot. The Clusters of Orthologous Groups of proteins (COG) database information was employed for the functional annotation of DEPs. The Worfpsort and Cello software packages were used to determine the subcellular localization of proteins of interest. Metabolic pathway analysis using the DEPs was performed via the KEGG database [21]. Finally, GO enrichment analysis was carried out to discover enriched pathways [22, 23].

## RNA isolation and real-time PCR

Sequences of 8 DEPs identified with iTRAQ were downloaded from the National Center for Biotechnology Information database. Total RNA was isolated from insects using TRIzol (Takara, Japan). cDNA was synthesized using a FastQuant RT Kit (TIANGEN), followed by qRT-PCR via the SuperReal PreMix Plus (Takara, Japan) under the following conditions: 95°C for 2 min; followed by 40 cycles of 95°C for 20 s, 60°C for 20 s, and 72°C for 30 s. The qRT-PCR analysis of each sample was carried out three times. Relative expression of each transcript was calculated via the $2^{-\Delta\Delta CT}$ method [24] and normalized with ß-Actin, with BPH-TN1 samples used as a negative control.

# Results

## Quantitative proteomic analysis by iTRAQ labeling

Proteins that were differentially expressed between BPH-YHY15 and BPH-TN1 were evaluated by iTRAQ labeling and quantified by liquid chromatography-mass spectrometry analysis (Fig 1). According to the LC-MS/MS analysis, we identified 3167 proteins in the two populations. Based on the criteria for defining DEPs (fold change ratio $\geq 1.2$ and $p \leq 0.05$), 151 up-regulated and 107 down-regulated proteins were discovered in the BPH-YHY15 population relative to the BPH-TN1 population (Fig 2). Information on the DEPs and their accession numbers are shown in the S1 Table.

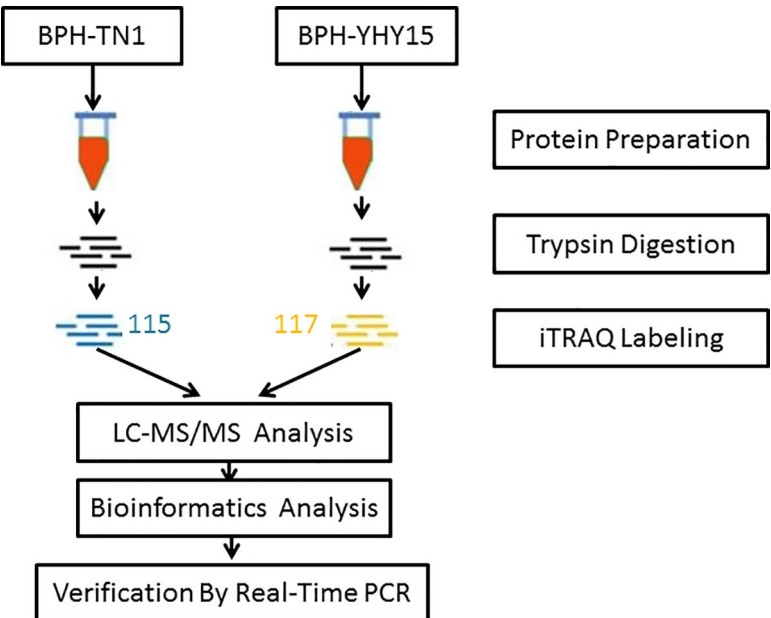

**Fig 1. Overview of the experimental workflow.** Differentially expressed proteins were quantified and analyzed by iTRAQ labeling and LC-MS/MS. A bioinformatics analysis was used to explore the relationship between differentially expressed proteins in *N. lugens* and BPH adaptation to rice resistance. qRT-PCR was used to demonstrate that protein abundance changes were correlated with differential expression at the transcriptional level.

## Functional annotation and classification

DEPs were queried against GO and KEGG databases. We were able to annotate the molecular functions of 293 proteins, including binding proteins and proteins with catalytic activity, transporter activity or structural importance (S2 Table, MF). For cellular components, we annotated 829 proteins present in either the cytosol or organelles (S2 Table, CC). Finally, 1055

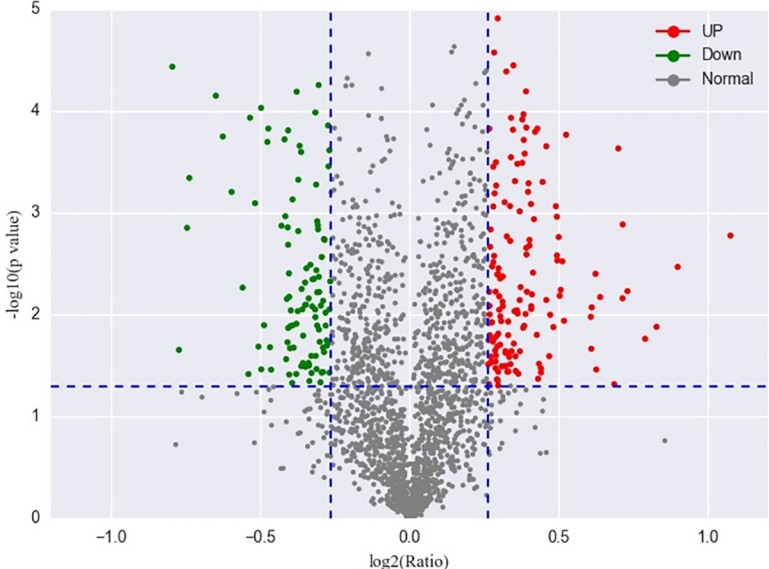

**Fig 2. Volcano plots of iTRAQ-labeled proteins.** The volcano plot shows up-regulated differentially expressed proteins (red), unchanged proteins (gray), and down-regulated differentially expressed proteins (green).

proteins were assigned to different biological processes, including cellular processes, single-organism processes and metabolic processes (S2 Table, BP).

## Gene Ontology (GO) enrichment analysis of DEPs

We next performed GO enrichment analysis using Fisher's exact test to identify processes that were enriched in the DEPs (S3 Table). Significantly enriched molecular function categories included lipid binding (11 proteins), carbon-carbon lyase activity (4), and alcohol binding (4) (Fig 3A). Cellular components were enriched in extracellular region (49), extracellular space (13), lytic vacuole (9) and lysosome (9) (Fig 3B). Biological process categories included response to chemical (38), response to oxygen-containing compound (20) and regulation of response to stress (19) (Fig 3C). Overall this analysis revealed that the DEPs were enriched in proteins that are responsive to chemicals.

## Kyoto Encyclopedia of Genes and Genomes (KEGG) pathway enrichment analysis

In addition to GO annotation, we also examined the enrichment of DEPs in different KEGG pathways. DEPs were found to be enriched in apoptosis (7 proteins) and lysosome metabolism (7 proteins), suggesting these processes may play critical roles in BPH adaptation to rice resistance (Fig 4, S4 Table). However, further studies of these pathways at the molecular and cellular levels should be performed to corroborate these findings.

## Clusters of Orthologous Groups of proteins (COG) pathway enrichment analysis of DEPs

We classified all DEPs with high homology into 25 COG categories (Fig 5; S5 Table). "Post-translational modification, protein turnover, chaperones" (31 proteins) was the most abundant functional category, followed by "general function prediction only" (22 proteins), and "energy production and conversion" (21 proteins). Additionally, many proteins were clustered in groups related to stress response, including "signal transduction mechanisms" (18 proteins), "inorganic ion transport and metabolism" (7 proteins), "transcription" (5 proteins), and "defense mechanisms" (3 proteins). Based on these clusters, we identified 20 DEPs correlated with brown planthopper adaptation to rice resistance from the 258 DEPs, likely lipid transport and metabolism (vitellogenin), secondary metabolites biosynthesis, transport and catabolism (cytochrome P450), amino acid transport and metabolism (serine protease HP21), defense mechanisms (serpin-4), and post-translational modification, protein turnover, chaperones (heat shock protein 70) (Table 1).

## Comparison of the proteomics data with transcriptional data

To investigate the relationship between the DEPs and their respective transcripts, we performed qRT-PCR on 8 expressed genes from the iTRAQ results (S6 Table), including four up-regulated (vitellogenin, proclotting enzyme-3, serpin-4 and cytochrome P450 CYP6ER1v2) (Fig 6A) and four down-regulated proteins (trypsin-26, beta-tubulin 4, chitinase and sugar transporter 1) (Fig 6B). They were chosen in conformity with the proportion of their up- and downregulation from the iTRAQ results and the availability of the mRNA sequence from the BPH transcriptome. Overall, the differential expression at the transcriptional level was well-correlated with the trends at the protein level, indicating that iTRAQ is a reliable method for differential protein analysis of BPH.

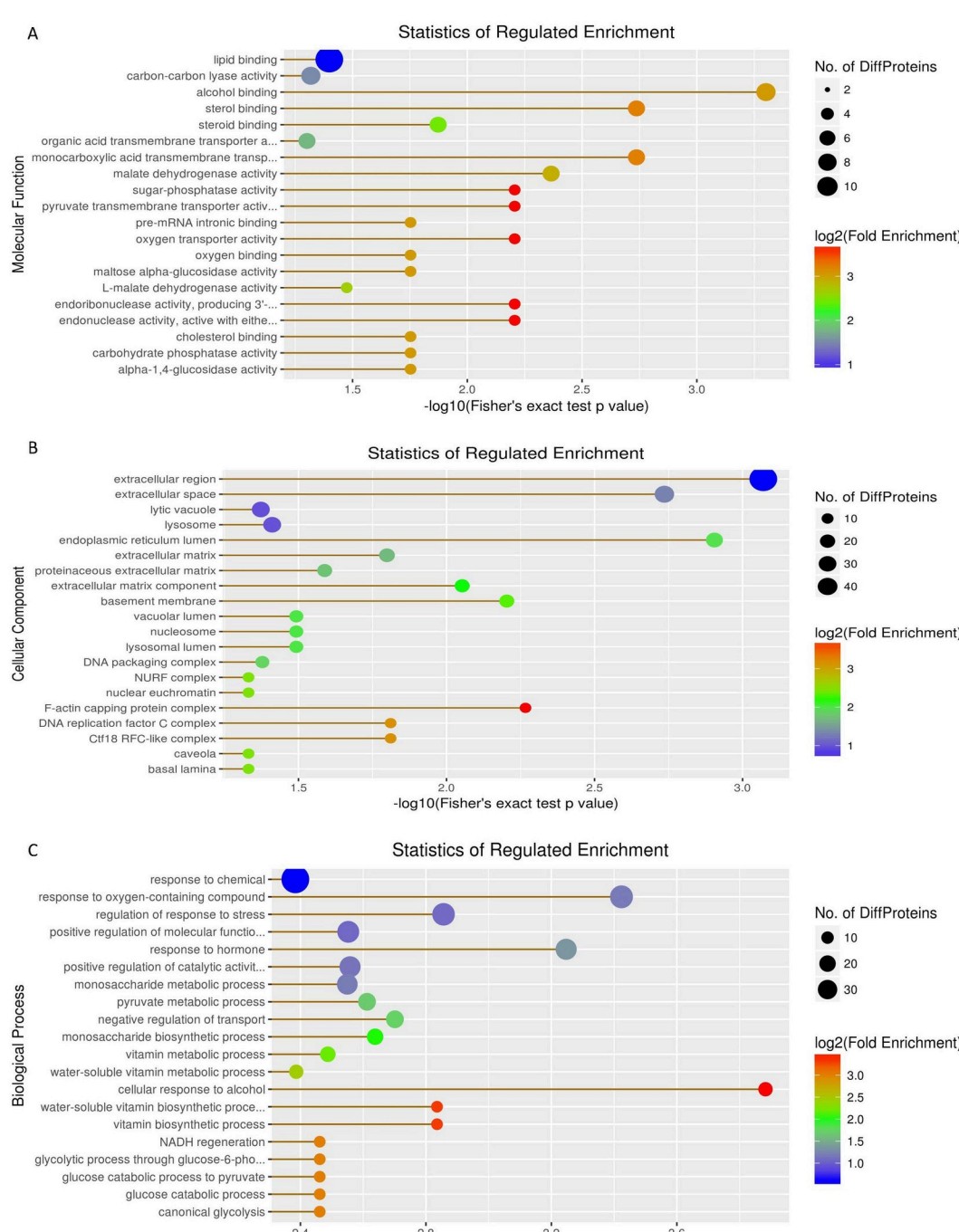

**Fig 3. Gene Ontology enrichment analysis of differentially expressed proteins identified in BPH-YHY15 and BPH-TN1.**
A: Molecular Function. B: Cellular Component. C: Biological Process. The X-axis represents Fisher's exact test p-value, while the Y-axis represents the name of the pathway. The circle color represents enrichment fold change, and the circle size represents the amount of differentially expressed proteins contained in each category.

## Discussion

iTRAQ is a new technique that identifies and quantifies proteins by using isobaric reagents to label the primary amines of peptides and proteins in a digestion mixture. S ions are used as the

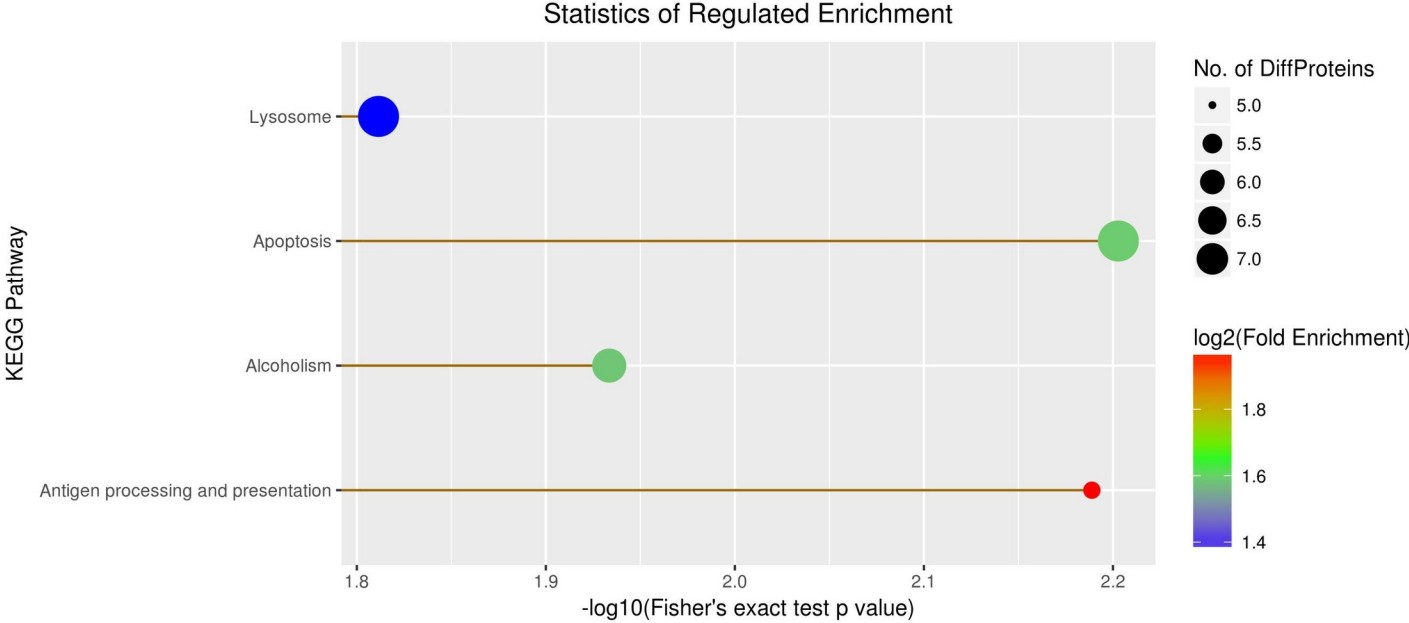

**Fig 4. Scatter diagram of enriched Kyoto Encyclopedia of Genes and Genomes pathways.** The X-axis represents Fisher's exact test p-value, while the Y-axis represents the name of the pathway. The circle color represents enrichment fold change, and the circle size represents the amount of differentially expressed proteins contained in each category.

reporter, which enables the quantification of proteins from different sources [25]. This method makes protein assays significantly higher throughput, leading to a significant increase in larger scale protein analyses [26, 27]. To determine the link between altered protein levels and adaptability in brown planthoppers exposed to susceptible and resistant rice varieties, iTRAQ was

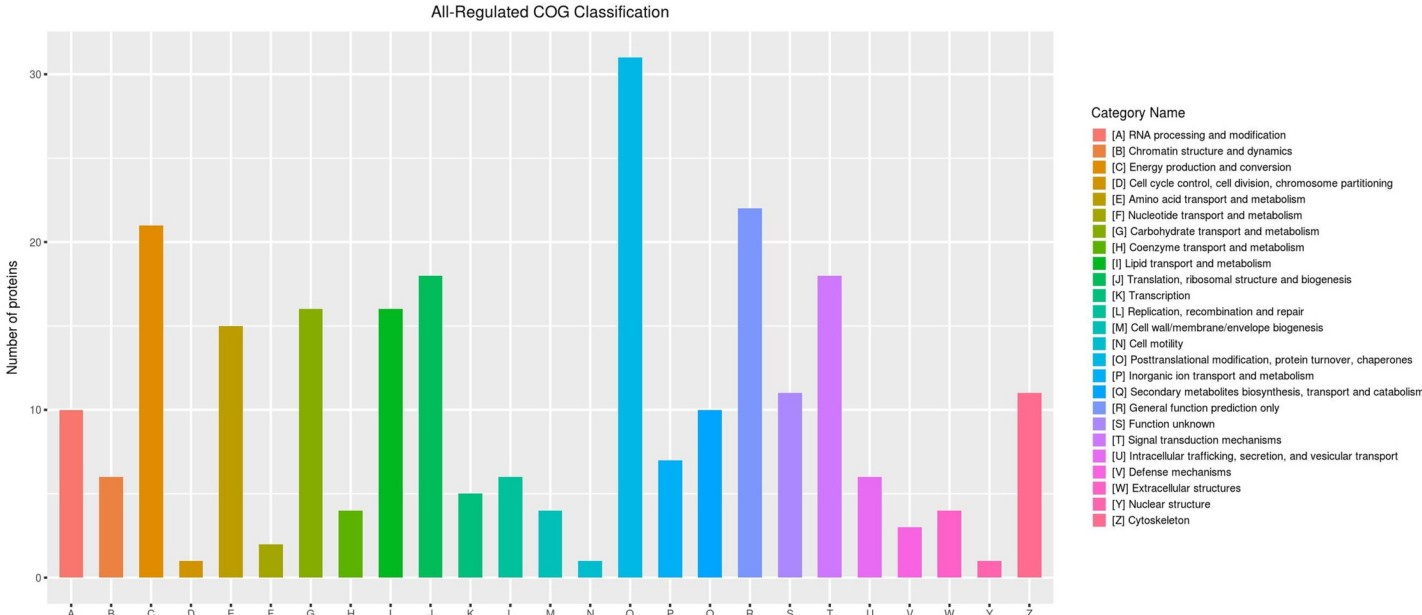

**Fig 5. Clusters of orthologous groups of proteins pathway enrichment analysis of differentially expressed proteins.** The X-axis represents classification, while the Y-axis represents the number of differentially expressed proteins.

**Table 1. DEPs associated with the adaptation of brown planthopper to resistant rice.**

| Accession | Description | Fold change | P-value |
|---|---|---|---|
| BAP87098.1 | vitellogenin | 2.11 | 0.00 |
| AGZ04899.1 | odorant binding protein 8 | 1.77 | 0.01 |
| AGK40915.1 | proclotting enzyme-3 | 1.41 | 0.00 |
| AID60312.1 | serine protease HP21 | 1.34 | 0.00 |
| AGK40928.1 | serpin-4 | 1.33 | 0.01 |
| AJO25056.1 | imaginal disc growth factor | 1.32 | 0.00 |
| AIW79986.1 | cytochrome P450 CYP6ER1v2 | 1.31 | 0.00 |
| AFJ75808.1 | glutathione s-transferase M2 | 1.30 | 0.00 |
| CAC87119.1 | trypsin-like protease | 1.21 | 0.03 |
| AKN21380.1 | multicopper oxidase 2 | 1.20 | 0.03 |
| AEL88646.1 | phosphoacetylglucosamine mutase | 0.83 | 0.00 |
| AID60360.1 | trypsin-26 | 0.82 | 0.01 |
| AIW79984.1 | cytochrome P450 CYP6CS1v2 | 0.81 | 0.01 |
| AMZ00358.1 | Ras superfamily small GTPase Cdc42 | 0.81 | 0.03 |
| AIW79998.1 | cytochrome P450 CYP380C10 | 0.80 | 0.04 |
| ALP82110.1 | beta-tubulin 4 | 0.78 | 0.01 |
| AJO25042.1 | chitinase, partial | 0.76 | 0.00 |
| CAZ65617.1 | cytochrome P450 | 0.74 | 0.00 |
| BAI83415.1 | sugar transporter 1 | 0.60 | 0.00 |
| AFJ20626.1 | heat shock protein 70 | 0.55 | 0.00 |

performed in combination with LC-MS/MS. This led to the identification of 3167 unique proteins, 258 of which were significantly differentially expressed between the two *N. lugens* populations. We then annotated differentially abundant proteins via GO analysis to determine their putative functions. As expected, several GO terms—including response to a stimulus, response to chemicals, regulation of response to stimulus, response to external stimulus, response to organic substance, cellular response to stress and regulation of cell communication—were found to be substantially enriched. Notably, a large number of the DEPs may be related to response to stimulus.

GO enrichment pathway analysis indicated that response to chemical, response to oxygen-containing compound, and regulation of response to stress related DEPs might be involved in

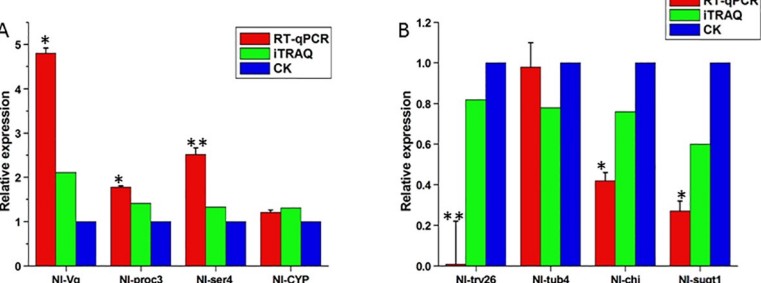

**Fig 6. Correlation between iTRAQ results and qRT-PCR results.** β-actin was used to normalize gene expression changes. Error bars represent SD, and all analyses were performed in triplicate. * $p < 0.05$, ** $p < 0.01$ (Student's t-test). A: Four genes (*Nl-Vg*, *Nl-proc3*, *Nl-ser4* and *Nl-CYP*) were found to be up-regulated at both the protein and transcriptional levels. B: Four genes (*Nl-try26*, *Nl-tub4*, *Nl-chi* and *Nl-sugt1*) were found to be down-regulated at both levels. Red and green represent the expression levels of BPH-YHY15 by qRT-qPCR and iTRAQ labeled techniques, respectively. CK represents that of BPH-TN1 for negative control.

BPH adaptation to rice resistance. Furthermore, KEGG enrichment pathway analysis identified that the DEPs were involved in specific metabolic pathways, including apoptosis metabolism. The result suggested that feeding on moderately resistant plants may induce cell apoptosis in BPH. Apoptosis is an effective means by which a host controls virus infection [28]. *Rice ragged stunt virus* will induce apoptosis in the salivary gland cells of its insect vector, *N. lugens* [29]. The apoptosis metabolism may be associated with BPH adaptation to rice resistance, and understanding the nature of these proteins can improve mechanistic understanding of resistance adaptation.

The COG pathway enrichment analysis showed that the largest group (12% of the DEPs) was associated with "post-translational modification, protein turnover, chaperones", such as heat shock protein 70. And we also found several DEPs correlated with brown planthopper adaptation to rice resistance likely lipid transport and metabolism (vitellogenin), secondary metabolites biosynthesis, transport and catabolism (cytochrome P450), and amino acid transport and metabolism (serine protease HP21).

Heat shock 70 kDa proteins (Hsp70) participate in protein folding, as well as protein translocation and caspase-dependent apoptosis [30]. Our results indicated that Hsp70 protein was more abundant in the BPH-YHY15 group, which may increase this group's survival rate at low temperatures. Relatively high levels of *hsp70* mRNA are correlated with low temperature tolerance in some insects, such as *D. melanogaster* [31] and *Leptinotarsa decemlineata* [32].

Vitellogenin is a precursor protein present in the egg yolk of many oviparous insects. However, insect Vgs are usually synthesized within body fat and processed with many co- and post-translational modifications, including proteolytic cleavage of nascent proteins [33–35]. Previous studies found Vg of *Apis mellifera* can protect itself from oxidative stress [36]. Moreover, the expression of *Nlvg* is correlated with the embryonic development stage in *N. lugens* [37]. In our study, we found that *Nlvg* expression in BPH-YHY15 individuals was 5.84-fold higher than that of the individuals in the BPH-TN1 population, suggesting that higher lipid deposition ability may be involved in the adaptation response of *N. lugens* to resistant rice.

Insect detoxification systems often evolve during insect-plant interactions via modification of P450s and glutathione S-transferases that catabolize secondary plant compounds [38–40]. P450s play crucial roles in insect growth and development and are also involved in pesticide metabolism [41]. Several studies have demonstrated that mRNA transcript levels of the BPH P450 gene were induced by some resistant rice varieties [42–44]. Based on our iTRAQ results, we infer that P450 enzymes may be involved in BPH adaptation to the rice YHY15.

Serine proteases are involved in insect digestion and play crucial roles in numerous physiological processes, including larval growth and development [45]. Higher expression levels of serine proteases in host-adapted BPHs suggest that they may offer nutrients required for BPH survival and fecundity.

In summary, iTRAQ led to the detection of 258 DEPs between the BPH-TN1 and BPH-YHY15 groups of BPH. In the DEPs, 151 were up-regulated and 107 were down-regulated. Determining how protein contents differ between host-adapted and non-adapted BPH populations can lead to an improved understanding of the adaptability of BPH to rice resistance at the proteomic level. Moreover, this data may also help us explain other BPH traits, including fecundity and survival. Despite these interesting findings, our study does have some significant limitations. In particular, some DEPs could not be identified and well-quantified, motivating future experimental work on this species. Taken together, our analysis of differences in protein abundance in the two virulent *N. lugens* populations represents a significant step forward in the understanding of BPH adaptability to resistant rice.

## Supporting information

**S1 Table. Differentially expressed proteins between BPH-YHY15 and BPH-TN1 samples.**
(XLSX)

**S2 Table. Functional annotation and classification of proteins.**
(XLSX)

**S3 Table. Functional enrichment in differentially expressed proteins.**
(XLSX)

**S4 Table. Significantly enriched KEGG pathways in differentially expressed proteins.**
(XLSX)

**S5 Table. Significantly enriched COG pathways in differentially expressed proteins.**
(XLSX)

**S6 Table. List of genes selected for qRT-PCR assay.**
(XLSX)

## Acknowledgments

We would like to thank TopEdit (www.topeditsci.com) for the English language editing of this manuscript.

## Author Contributions

**Conceptualization:** Wenjun Zha.

**Data curation:** Wenjun Zha.

**Project administration:** Aiqing You.

**Writing – original draft:** Wenjun Zha.

**Writing – review & editing:** Aiqing You.

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
