## [Decision Letter · Decision Letter 0]

26 Jun 2020

PONE-D-20-16306

iTRAQ-based quantitative proteomic analysis of proteins associated with the adaptation of brown planthopper to resistant rice varieties

PLOS ONE

Dear Dr. You,

Thank you for submitting your manuscript to PLOS ONE. After careful consideration, we feel that it has merit but does not fully meet PLOS ONE’s publication criteria as it currently stands. Therefore, we invite you to submit a revised version of the manuscript that addresses the points raised during the review process.

We look forward to receiving your revised manuscript.

Kind regards,

Peng He, Ph.D

Academic Editor

PLOS ONE

Journal Requirements:

Reviewers' comments:

Reviewer's Responses to Questions

**Comments to the Author**

1. Is the manuscript technically sound, and do the data support the conclusions?

Reviewer #1: Partly

Reviewer #2: Yes

2. Has the statistical analysis been performed appropriately and rigorously? 

Reviewer #1: Yes

Reviewer #2: Yes

3. Have the authors made all data underlying the findings in their manuscript fully available?

Reviewer #1: Yes

Reviewer #2: Yes

4. Is the manuscript presented in an intelligible fashion and written in standard English?

Reviewer #1: Yes

Reviewer #2: Yes

5. Review Comments to the Author

Reviewer #1: The manuscript contributed by Zha et al. investigated related proteins contributed to the adaptation of Nilaparvata lugens to resistant rice varieties by iTRAQ. Authors found differentially expressed proteins (DEPs) between two virulent N. lugens, which were associated with lipid transport and metabolism and defense mechanism. Authors concluded that these DEPs may participate in N. lugens adaptation to resistant rice varieties. However, there are several points or things to be revised or answered.

1. In Short Title, authors should indicate what proteins to be analyzed or what physiological problems to be solved by iTRAQ.

2. The title of this paper is very general, please concretize this title.

3. From line 12 to line 13, authors didn't resolve how the proteins of N. lugens mediate its adaptation to rice resistance throughout the whole paper, only some differentially expressed proteins were identified by iTRAQ. Please revise this sentence expression.

4. From line 14 to line 15, "Biotype Y" and "Biotype I" didn't appear in Materials and Methods. Why were they here? Please explain and revise.

5. From line 19 to line 21, whether the protein expression level can be detected by qRT-PCR? Please make clear this problem.

6. From line 21 to line 23, authors should consider to rewrite this research conclusion. Lipid transport and metabolism and defense mechanism were not showed in KEGG pathway enrichment analysis (3.4), and number of proteins of the two categories is not the most compared with others (3.5). Please explain and revise.

7. In introduction, authors need to introduce reason of selecting YHY15 and TN1.

8. In second paragraph of introduction, the contents of pest resistant rice varieties cultivation and the adaptation mechanism of N. lugens to resistant rice need to be added, which can let readers kown the progress of research about pest resistant rice. Please revise.

9. In third paragraph of introduction, the main research contents of this paper are drafted according to Materials and Methods and Results. Please authors consider to reorganize this part.

10. In Results, the top 20 or 30 differentially expressed proteins (DEPs) between two virulent N. lugens should be considered to be added as table, which can better understand the adaptation mechanism of N. lugens to resistant rice. In this way, this paper can show specific DEPs associated with the adaptation of N. lugens to resistant rice varieties. Please revise.

11. In Discussion, vitellogenin, P450, serine proteases and Hsp70 didn't appear throughout paper Results, why did authors suddenly discuss these proteins? I suggest that discussion is written according to research results. Please reorganize article discussion.

12. In line 258, "Stal" is wrong. Please check similar problem.

Reviewer #2: This manuscript performed protein quantitation using iTRAQ and then compared the expression

patterns between two virulent N. lugens populations and found 258 differentially

expressed proteins. It was well organized and written, thus I recommend acceptance after minor revision.

Major concerns:

1. Two virulent N. lugens populations (BPH-TN1 and BPH-YHY15): should describe basic data of these 2 populations, for example, survival rates on YHY15 or TN1. Alternatively, related references could be cited.

2. 8 DEPs were chosen for qRT-PCR. How to select?

6. PLOS authors have the option to publish the peer review history of their article (what does this mean?). If published, this will include your full peer review and any attached files.

Reviewer #1: No

Reviewer #2: No

---

## [Author Response · Author response to Decision Letter 0]

18 Jul 2020

Response to Reviewer 1:

Comment 1: 

The manuscript contributed by Zha et al. investigated related proteins contributed to the adaptation of Nilaparvata lugens to resistant rice varieties by iTRAQ. Authors found differentially expressed proteins (DEPs) between two virulent N. lugens, which were associated with lipid transport and metabolism and defense mechanism. Authors concluded that these DEPs may participate in N. lugens adaptation to resistant rice varieties. However, there are several points or things to be revised or answered.

1. In Short Title, authors should indicate what proteins to be analyzed or what physiological problems to be solved by iTRAQ.

Answer: 

Thanks for the Reviewer’s valuable comments. The short title "iTRAQ-based quantitative proteomic analysis of brown planthopper proteins" was changed to "iTRAQ proteomic reveals the adaptation of brown planthopper to two rice varieties" in the revised manuscript.

Comment 2：

The title of this paper is very general, please concretize this title.

Answer: 

Following your suggestion, the title of this paper "iTRAQ-based quantitative proteomic analysis of proteins associated with the adaptation of brown planthopper to resistant rice varieties" was changed to "Comparative iTRAQ proteomic profiling of proteins associated with the adaptation of brown planthopper to moderately resistant vs. susceptible rice varieties" in the revised manuscript (Lines 1-2, Page 1).

Comment 3：

From line 12 to line 13, authors didn't resolve how the proteins of N. lugens mediate its adaptation to rice resistance throughout the whole paper, only some differentially expressed proteins were identified by iTRAQ. Please revise this sentence expression.

Answer: 

Following your suggestion, we changed " the proteins of N. lugens mediate its adaptation to rice resistance " to " BPHs adapt to the resistant rice variety " in the revised manuscript (Lines 12-13, Page 1).

Comment 4：

From line 14 to line 15, "Biotype Y" and "Biotype I" didn't appear in Materials and Methods. Why were they here? Please explain and revise.

Answer: 

We provided a brief description for Biotype Y and Biotype I in the abstract section, because it will help a person outside of this field to comprehend them well. And following your suggestion, we changed "BPH-TN1 and BPH-YHY15" to " Biotype I and Biotype Y " in Materials and Methods section of the revised manuscript (Line 71, Page 2).

Comment 5： 

From line 19 to line 21, whether the protein expression level can be detected by qRT-PCR? Please make clear this problem.

Answer: 

Following your suggestion, we changed " of these DEPs " to " expressed genes from the iTRAQ results " in the revised manuscript (Line 19, Page 1). And we also changed " selected DEPs " to " expressed genes from the iTRAQ results "(Line 175, Page 5).

Comment 6： 

From line 21 to line 23, authors should consider to rewrite this research conclusion. Lipid transport and metabolism and defense mechanism were not showed in KEGG pathway enrichment analysis (3.4), and number of proteins of the two categories is not the most compared with others (3.5). Please explain and revise.

Answer: 

Following your suggestion, we changed " Our evidence suggests that DEPs of N. lugens are associated with lipid transport and metabolism, as well as defense mechanisms, which may contribute to BPH adaptation to resistant rice varieties. " to " The determination of the protein changes in two virulent N. lugens populations would help to better understanding BPH adaptation to resistant rice varieties and facilitate better design of new control strategies for host defense against BPH. " in the revised manuscript (Lines 21-23, Page 1).

Comment 7： 

In introduction, authors need to introduce reason of selecting YHY15 and TN1.

Answer: 

Following your suggestion, we added the reason why these 2 populations were chosen. We added the paragraph “Among different brown planthopper biotypes, the BPH biotype I is widely distributed in East and Southeast Asia and can survive on the TN1 rice variety [16]. The BPH biotype Y is a virulent biotype that has adapted to the moderately resistant rice variety (YHY15) by compelling biotype I BPHs to feed on YHY15 for generations [17]. "into our revised manuscript and the details can be found in Lines 51-55, Page 2.

Comment 8： 

In second paragraph of introduction, the contents of pest resistant rice varieties cultivation and the adaptation mechanism of N. lugens to resistant rice need to be added, which can let readers kown the progress of research about pest resistant rice. Please revise.

Answer: 

Following your suggestion, we added the contents of pest resistant rice varieties cultivation and the adaptation mechanism of N. lugens to resistant rice. We added the paragraph “Since the first resistant rice variety against BPH was discovered in 1969, more than 30 BPH resistance genes have been reported from different resistance sources [11]. We used a susceptible rice variety (TN1) as a control and a moderately resistant rice variety (YHY15) carrying the resistance gene BPH15 [12]. It has been found that resistance genes impair BPH feeding behavior on varieties and cause BPH physiological changes by increasing mortality rates, extending developmental periods, and reducing reproductive output [10, 13, 14]. BPHs that are allowed to feed on resistant rice for a long time may slowly evolve into new virulent BPH populations to overcome rice resistance [15]. " into our revised manuscript and the details can be found in Lines 44-51, Page 2.

Comment 9： 

In third paragraph of introduction, the main research contents of this paper are drafted according to Materials and Methods and Results. Please authors consider to reorganize this part.

Answer: 

Following your suggestion, we reorganized this part according to Results. The paragraph is “In this study, iTRAQ was used to evaluate proteomic differences between two BPH populations, leading to the identification of DEPs which are correlated with resistance. Among 3167 identified proteins, 258 were considered as differentially expressed in the BPH-YHY15 population relative to the BPH-TN1 population. We then used Gene Ontology (GO) annotations and Kyoto Encyclopedia of Genes and Genomes (KEGG) pathway analysis to analyze the functions of these DEPs. Subsequent Clusters of Orthologous Groups of proteins (COG) analysis suggested that a number of those proteins were involved in the regulation of post-translational modification, protein turnover, chaperones pathways. Additional research on these proteins might provide valuable information regarding strategies for BPH management and control.” in the revised manuscript (Lines 57-66, Page 2).

Comment 10： 

In Results, the top 20 or 30 differentially expressed proteins (DEPs) between two virulent N. lugens should be considered to be added as table, which can better understand the adaptation mechanism of N. lugens to resistant rice. In this way, this paper can show specific DEPs associated with the adaptation of N. lugens to resistant rice varieties. Please revise.

Answer: 

Following your suggestion, we added specific DEPs associated with the adaptation of N. lugens to resistant rice varieties in Table 1 of our revised manuscript. And information on the DEPs and their accession numbers are shown in S1 Table.

Comment 11： 

In Discussion, vitellogenin, P450, serine proteases and Hsp70 didn't appear throughout paper Results, why did authors suddenly discuss these proteins? I suggest that discussion is written according to research results. Please reorganize article discussion.

Answer: 

Following your suggestion, we added vitellogenin, P450, serine proteases and Hsp70 into the results of our revised manuscript (Lines 166-171, Page 5). The paragraph is “Based on these clusters, we identified 20 DEPs correlated with brown planthopper adaptation to rice resistance from the 258 DEPs, likely Lipid transport and metabolism (vitellogenin), Secondary metabolites biosynthesis, transport and catabolism (cytochrome P450), Amino acid transport and metabolism (serine protease HP21), Defense mechanisms (serpin-4), and Posttranslational modification, protein turnover, chaperones (heat shock protein 70) （Table 1）.”

Comment 12： 

In line 258, "Stal" is wrong. Please check similar problem.

Answer: 

Following your suggestion, we have checked all the similar problems.

Response to Reviewer 2:

Comment 1： 

Two virulent N. lugens populations (BPH-TN1 and BPH-YHY15): should describe basic data of these 2 populations, for example, survival rates on YHY15 or TN1. Alternatively, related references could be cited.

Answer: 

Following your suggestion, we added the reason why these 2 populations were chosen. We added the paragraph " Since the first resistant rice variety against BPH was discovered in 1969, more than 30 BPH resistance genes have been reported from different resistance sources [11]. We used a susceptible rice variety (TN1) as a control and a moderately resistant rice variety (YHY15) carrying the resistance gene BPH15 [12]. It has been found that resistance genes impair BPH feeding behavior on varieties and cause BPH physiological changes by increasing mortality rates, extending developmental periods, and reducing reproductive output [10, 13, 14]. BPHs that are allowed to feed on resistant rice for a long time may slowly evolve into new virulent BPH populations to overcome rice resistance [15]. Among different brown planthopper biotypes, the BPH biotype I is widely distributed in East and Southeast Asia and can survive on the TN1 rice variety [16]. The BPH biotype Y is a virulent biotype that has adapted to the moderately resistant rice variety (YHY15) by compelling biotype I BPHs to feed on YHY15 for generations [17]. " into our revised manuscript and the details can be found in Lines 44-55, Page 2.

Comment 2： 

8 DEPs were chosen for qRT-PCR. How to select?

Answer: 

Following your suggestion, we added the reason why 8 DEPs were chosen for qRT-PCR. We added the sentence" They were chosen in conformity with the proportion of their up- and downregulation from the iTRAQ results and the availability of the mRNA sequence from the BPH transcriptome." into our revised manuscript and the details can be found in Lines 178-180, Page 5.

---

## [Decision Letter · Decision Letter 1]

6 Aug 2020

PONE-D-20-16306R1

Comparative iTRAQ proteomic profiling of proteins associated with the adaptation of brown planthopper to moderately resistant vs. susceptible rice varieties

PLOS ONE

Dear Dr. You,

Thank you for submitting your manuscript to PLOS ONE. After careful consideration, we feel that it has merit but does not fully meet PLOS ONE’s publication criteria as it currently stands. Therefore, we invite you to submit a revised version of the manuscript that addresses the points raised during the review process.

We look forward to receiving your revised manuscript.

Kind regards,

Peng He, Ph.D

Academic Editor

PLOS ONE

Reviewers' comments:

Reviewer's Responses to Questions

**Comments to the Author**

1. If the authors have adequately addressed your comments raised in a previous round of review and you feel that this manuscript is now acceptable for publication, you may indicate that here to bypass the “Comments to the Author” section, enter your conflict of interest statement in the “Confidential to Editor” section, and submit your "Accept" recommendation.

Reviewer #1: (No Response)

Reviewer #2: All comments have been addressed

2. Is the manuscript technically sound, and do the data support the conclusions?

Reviewer #1: Yes

Reviewer #2: Yes

3. Has the statistical analysis been performed appropriately and rigorously? 

Reviewer #1: No

Reviewer #2: Yes

4. Have the authors made all data underlying the findings in their manuscript fully available?

Reviewer #1: Yes

Reviewer #2: Yes

5. Is the manuscript presented in an intelligible fashion and written in standard English?

Reviewer #1: No

Reviewer #2: Yes

6. Review Comments to the Author

Reviewer #1: The manuscript contributed by Zha et al. characterized proteomic profiling of proteins associated with the adaptation of brown planthopper to resistant rice varieties by iTRAQ. Authors found differentially expressed proteins (DEPs) between two virulent N. lugens, which would help to better understanding BPH adaptation to resistant rice varieties and facilitate better design of new control strategies for host defense against BPH. Although authors revised several points according to the academic editor and reviewers suggestions, there are several points or things to be revised or answered.

1. In Keywords section, the selection of keywords should be consistent with professional research content, please add keyword.

2. The quality of the written English requires improvement for better.

(1) In line 80, a space is required in "-80°C". Please check for similar problems in this paper.

(2) In line 93, a space is required in "5%ACN". Please check for similar problems in this paper.

(3) "Nano-LC-MS/MS" of line 92 and "Nano LC-MS/MS" of line 98 are not consistent.

(4) Please revise font size in line 166-171. Please check for similar problems in the whole paper.

3. Please add statistical analysis in Figure 6.

4. In Discussion, authors should add some discussion contents about GO enrichment analysis of DEPs, KEGG pathway enrichment analysis of DEPs, COG pathway of DEPs. Vitellogenin, P450, serine proteases and Hsp70 should be integrated into pathway to discuss, Please consider whether any changes are needed. In addition, reference 36 and 37 are about insecticide resistance, which are not suitable here, please revise.

Reviewer #2: The authors addressed all comments raised by the reviewers, so I have no more questions. Thus, I suggest it be accepted for publication in the journal.

7. PLOS authors have the option to publish the peer review history of their article (what does this mean?). If published, this will include your full peer review and any attached files.

Reviewer #1: No

Reviewer #2: No

---

## [Author Response · Author response to Decision Letter 1]

16 Aug 2020

Response to Reviewer 1:

Comment 1: 

The manuscript contributed by Zha et al. characterized the proteomic profiling of proteins associated with the adaptation of brown planthopper to resistant rice varieties by iTRAQ. Authors found differentially expressed proteins (DEPs) between two virulent N. lugens, which would help to better understanding BPH adaptation to resistant rice varieties and facilitate the better design of new control strategies for host defense against BPH. Although authors revised several points according to the academic editor and reviewers suggestions, there are several points or things to be revised or answered.

1. In the Keywords section, the selection of keywords should be consistent with professional research content, please add the keyword.

Answer: 

Thanks for the Reviewer’s valuable comments. We added the keywords "adaptation; GO analysis; KEGG pathway analysis; COG pathway analysis" into the revised manuscript.

Comment 2：

The quality of written English requires improvement for better.

(1) In line 80, a space is required in "-80°C". Please check for similar problems in this paper.

Answer: 

Following your suggestion, "-80°C" was changed to "-80 °C" in the revised manuscript (Line 80, Page 2). And we have checked all the similar problems.

 (2) In line 93, a space is required in "5%ACN". Please check for similar problems in this paper.

Answer: 

Following your suggestion, a space was added into "5%ACN" in the revised manuscript (Line 93, Page 3).

(3) "Nano-LC-MS/MS" of line 92 and "Nano LC-MS/MS" of line 98 are not consistent.

Answer: 

Following your suggestion, "Nano LC-MS/MS" was changed to "Nano-LC-MS/MS" in the revised manuscript (Line 98, Page 3). 

(4) Please revise the font size in lines 166-171. Please check for similar problems in the whole paper.

Answer: 

Following your suggestion, we revised the font size in lines 166-171, Page 5.

Comment 3： 

Please add statistical analysis in Figure 6.

Answer: 

Following your suggestion, we added the statistical analysis (* p < 0.05, ** p < 0.01) into the Figure 6. 

Comment 4： 

In Discussion, authors should add some discussion contents about GO enrichment analysis of DEPs, KEGG pathway enrichment analysis of DEPs, COG pathway of DEPs. Vitellogenin, P450, serine proteases and Hsp70 should be integrated into the pathway to discuss. Please consider whether any changes are needed. In addition, reference 36 and 37 are about insecticide resistance, which are not suitable here, please revise.

Answer: 

Following your suggestion, first, we added some discussion contents about GO enrichment analysis of DEPs, KEGG pathway enrichment analysis of DEPs, COG pathway of DEPs in the revised manuscript. The paragraph is, “GO enrichment analysis indicated that response to chemical, response to oxygen-containing compound, and regulation of response to stress related DEPs might be involved in BPH adaptation to rice resistance. Furthermore, KEGG pathway enrichment analysis identified that the DEPs were involved in specific metabolic pathways, including apoptosis metabolism. The result suggested that feeding on moderately resistant plants may induce cell apoptosis in BPH. Apoptosis is an effective means by which a host controls virus infection [28]. Rice ragged stunt virus will induce apoptosis in the salivary gland cells of its insect vector, N. lugens [29]. The apoptosis metabolism may be associated with BPH adaptation to rice resistance, and understanding the nature of these proteins can improve mechanistic understanding of resistance adaptation.COG pathway enrichment analysis showed that the largest group (12% of the DEPs) was associated with "post-translational modification, protein turnover, chaperones", such as heat shock protein 70. And we also found several DEPs correlated with brown planthopper adaptation to rice resistance, likely lipid transport and metabolism (vitellogenin), secondary metabolites biosynthesis, transport and catabolism (cytochrome P450), and amino acid transport and metabolism (serine protease HP21).” (Lines 199-214, Pages 5-6).

Second, vitellogenin, P450, serine proteases and hsp70 were integrated into the COG pathway in the discussion part of the revised manuscript. 

Last, we deleted the reference 36 and 37 in the revised manuscript.

---

## [Editor Report · Decision Letter 2]

19 Aug 2020

Comparative iTRAQ proteomic profiling of proteins associated with the adaptation of brown planthopper to moderately resistant vs. susceptible rice varieties

PONE-D-20-16306R2

Dear Dr. You,

We’re pleased to inform you that your manuscript has been judged scientifically suitable for publication and will be formally accepted for publication once it meets all outstanding technical requirements.

Kind regards,

Peng He, Ph.D

Academic Editor

PLOS ONE
---

## [Editor Report · Acceptance letter]

24 Aug 2020

PONE-D-20-16306R2 

Comparative iTRAQ proteomic profiling of proteins associated with the adaptation of brown planthopper to moderately resistant vs. susceptible rice varieties 

Dear Dr. You:

I'm pleased to inform you that your manuscript has been deemed suitable for publication in PLOS ONE. Congratulations! Your manuscript is now with our production department. 

Kind regards, 

on behalf of

Dr. Peng He 

Academic Editor

PLOS ONE